# Causal Discovery Using Proxy Variables

**Mateo Rojas-Carulla**
Facebook AI Research, Paris
University of Cambridge
Max Plank Institute for Intelligent Systems, Tübingen
mateo.rojas-carulla@tuebingen.mpg.de

**Marco Baroni & David Lopez-Paz**
Facebook AI Research, Paris
{marco.baroni, dlp}@fb.com

## Abstract

In this paper, we develop a framework to estimate the cause-effect relation between two *static entities* $x$ and $y$: for instance, an art masterpiece $x$ and its fraudulent copy $y$. To this end, we introduce the notion of *proxy variables*, which allow the construction of a pair of *random entities* $(A, B)$ from the pair of static entities $(x, y)$. Then, estimating the cause-effect relation between $A$ and $B$ using an observational causal discovery algorithm leads to an estimation of the cause-effect relation between $x$ and $y$. We evaluate our framework in vision and language.

## 1 Introduction

Over the last decade, the state-of-the-art in *observational causal discovery* has matured into a wide array of algorithms (Shimizu et al., 2006; Hoyer et al., 2009; Daniusis et al., 2012; Peters et al., 2014; Mooij et al., 2016; Lopez-Paz et al., 2015; Lopez-Paz, 2016). All these algorithms estimate the causal relations between the random variables $(X_1, \ldots, X_p)$ by estimating various asymmetries in $P(X_1, \ldots, X_p)$. These methods consider *random entities* $X$ and $Y$, but often we are interested instead in two *static entities* $x$ and $y$. These are a pair of single objects for which it is not possible to define a probability distribution directly. Examples of such static entities may include one art masterpiece and its fraudulent copy, one translated document and its original version, or one pair of causally linked concepts in natural language, such as "virus" and "death".

**Our Contributions:** First, we introduce the framework of *proxy variables* to estimate the causal relation between *static entities* (Section 2). Second, we apply our framework to the task of inferring the cause-effect relation between pairs of images. In particular, our methods are able to infer the causal relation between an image and its stylistic modification in 72% of the cases, and it can recover the correct ordering of a set of shuffled video frames (Section 3). Third, we apply our framework to discover the cause-effect relation between pairs of concepts in natural language (Section 4). To this end, we introduce a novel dataset of 10,000 human-elicited pairs of words with known causal relation. Our methods are able to recover 75% of these challenging cause-effect relations.

## 2 Static Entities, Proxy Variables and Proxy Projections

We consider two *static entities* $x, y$ in some space $\mathcal{S}$ that satisfy the relation "$x$ causes $y$". This causal relation manifests the existence of a (possibly noisy) mechanism $f$ such that the value $y$ is computed as $y \leftarrow f(x)$. This asymmetric assignment guarantees changes in $x$ would lead to changes in $y$, but the converse would not hold. The two ingredients of our framework are: i) a *proxy random variable* $W$ is a random variable taking values in some set $\mathcal{W}$, which can be understood as a random source of information related to $x$ and $y$ (specific examples will be given in the different applications) and ii) a *proxy projection* is a function $\pi : \mathcal{W} \times \mathcal{S} \to \mathbb{R}$. Using proxy and projection, we can construct a pair of scalar random variables $A = \pi(W, x)$ and $B = \pi(W, y)$. The process is summarized in Figure 1 (right). A proxy variable and projection are *causal* if the pair of random entities $(A, B)$ share the same *causal footprint* as the pair of static entities $(x, y)$.[1] If the proxy variable and projection are causal, we may estimate the cause-effect relation between the static entities $x$ and $y$ in three steps. First, draw $(a_1, b_1), \ldots, (a_n, b_n)$ from $P(A, B)$. Second, use an observational causal discovery algorithm

---

[1] The concept of causal footprint is relative to our assumptions. For instance, when assuming an ANM $Y \leftarrow f(X) + N$, the causal footprint is the statistical independence between $X$ and $N$.

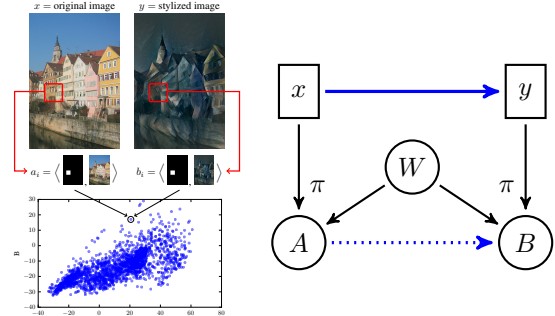

Figure 1: Left: Sampling random patches produces a proxy variable. Right: The *static entities* $(x, y)$ share a causal relation (thick blue arrow). A *proxy variable* $W$, and a *proxy projection* $\pi$ produce the *random entities* $(A, B)$, who share the causal footprint of $(x, y)$ (dotted blue arrow). Original image by Andreas Praefcke, CC BY 3.0.

| Experiment | Laplace | Prewitt | Artistic |
|---|---|---|---|
| ImageNet, Laplace | 70% | 70% | 72% |

Table 1: Performance of ANM on detecting the original image from its modified version.

to estimate the cause-effect relation between $A$ and $B$ given $\{(a_i, b_i)\}_{i=1}^n$. Third, conclude "$x$ causes $y$" if $A \rightarrow B$, or "$y$ causes $x$" if $A \leftarrow B$. Our code will be available soon.

## 3 CAUSAL DISCOVERY USING PROXIES IN IMAGES

Consider the two images shown in Figure 1 (left). The image on the left is an unprocessed photograph, while the one on the right is the same photograph after being stylized with the algorithm of Gatys et al. (2016). From a causal point of view, the unprocessed image $x$ is the cause of the stylized image $y$. How can we recover such causal relation using a standard causal discovery method?

The following is one possible solution. Assume that the two images are represented by pixel intensity vectors $x$ and $y$, respectively. For $n \gg 1$ and $j = 1, \ldots, n$: 1) Draw a mask-image $w_j$, which contains ones inside a patch at random coordinates, and zeroes elsewhere and 2) compute $a_j = \langle w_j, x \rangle$, and $b_j = \langle w_j, y \rangle$. In other words, $a_j$ and $b_j$ are the sums of pixel intensities of patch $w_j$ in images $x$ and $y$ respectively. This process returns a sample $\{(a_j, b_j)\}_{j=1}^n$ drawn from $P(A, B)$, the joint distribution of the two scalar random variables $(A, B)$. The conversion from static entities $(x, y)$ to random variables $(A, B)$ is obtained by virtue of i) the randomness generated by the *proxy variable* $W$, in this case random masks and ii) a causal projection $\pi$, here a simple dot product.

We conducted three experiments on images: i) we applied Laplace, Prewitt and Fourier filter to 1000 ImageNet images, ii) we downloaded 2000 original and stylized image pairs made using the algorithm of Gatys et al. (2016) available at `https://deepart.io` and iii) we also decompose a video of drops of ink mixing with water into 8 frames $\{(x_i)\}_{i=1}^8$, shown in Figure 2. Using the same mask proxy variable, we construct an $8 \times 8$ matrix $M$ such that $M_{ij} = 1$ if $x_i \rightarrow x_j$ according to our method and $M_{ij} = 0$ otherwise. Then, we consider $M$ to be the adjacency matrix of the DAG describing the causal structure between the 8 frames. By employing topological sort on this graph, we obtain the unique true ordering among $40,320$ possibilities. The results can be found in Table 1.

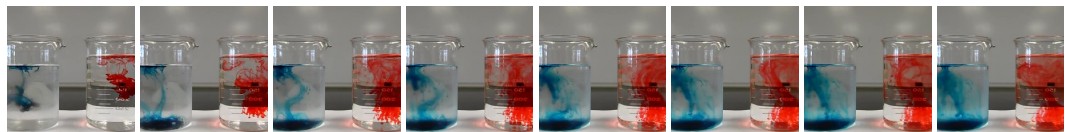

Figure 2: Proxy variables uncover the causal signal to reorder a shuffled sequence of video frames. The original video can be found in `http://goo.gl/sqdvu1`.

## 4 CAUSAL DISCOVERY USING PROXIES IN LANGUAGE

In the language of causal discovery with proxies, a pair of concepts denoted by words is a pair of static entities: $(x, y) = (\text{virus}, \text{death})$. We use a simple proxy: let $P(W = w)$ be the probability

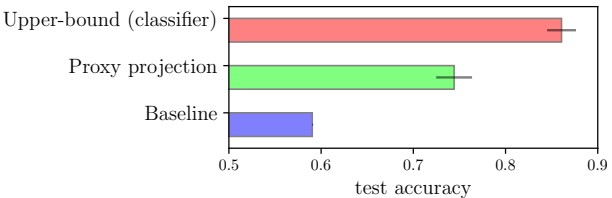

Figure 3: Results for best methods on the NLP experiment. Accuracies above $52\%$ are statistically significant with respect to a Binomial test at a significance level $\alpha = 0.05$.

of the word $w$ appearing in a sentence drawn at random from a large corpus of natural language. Using the proxy $W$, we define the pair of random variables $A = \pi_{\text{w2voi}}(W, x)$ and $B = \pi_{\text{w2voi}}(W, y)$, with $\pi_{\text{w2voi}}(w, x) = \langle v_w^o, v_x^i \rangle$, where $v_z^o \in \mathbb{R}^d$ and $v_z^i \in \mathbb{R}^d$ are the *output* and *input* word2vec representations (Mikolov et al., 2013) of the word $z$. The dot-product $\langle v_w^o, v_x^i \rangle$ measures the *similarity in context* between $w$ and $x$. We experimented with other projections, but do not report results as they were not statistically distinguishable, see Appendix D. Given this projection, we can sample $w_1, \ldots, w_n \sim P(W)$, compute the projections and perform causal discovery on $\{(a_i, b_i)\}_{i=1}^n$.

We introduce a human-elicited, human-filtered dataset of $10,000$ pairs of words with a known causal relation constructed by workers from Amazon Mechanical Turk. Details are given in Appendix A. We study methods that fall within three categories: *baselines*, *proxy projection methods*, and *feature-based supervised methods*. These three families of methods consider an increasing amount of information about the task at hand, and therefore exhibit an increasing performance up to $85\%$ classification accuracy. We report the best method in each family in Figure 3, and the full results in Appendix D. The best performing baseline is *precedence*: considering only sentences from the corpus in which both $x$ and $y$ appear, we predict "$x$ causes $y$" if $x$ precedes $y$ in more than half those sentences.

**Proxy projection methods:** Given $N$ word pairs $(x_i, y_i)$, this family of methods constructs a dataset $D = \left\{ (\{(a_j^i, b_j^i)\}_{j=1}^n, \ell^i) \right\}_{i=1}^N$, where $a_j^i = \pi_{\text{w2voi}}(w_j, x_i)$, $b_j^i = \pi_{\text{w2voi}}(w_j, y_i)$, $\ell^i = +1$ if $x_i \to y_i$ and $\ell^i = -1$ otherwise. We first split the dataset $D$ into a training set $D_{\text{tr}}$ with $75\%$ of the pairs and a test set $D_{\text{te}}$ with the remaining pairs. Then, the methods train RCC on the training set $D_{\text{tr}}$, and test its classification accuracy on $D_{\text{te}}$. This process is repeated ten times at random.

**Upper-bound for reference:** As an upper-bound on achievable test performance, we treat the same data as fixed-size vectors fed to a generic classifier, rather than random samples to be analyzed with a causal discovery method. This provides an **upper bound** on test performance, and can be seen as an oracle to the amount of causal signals (and signals correlated to causality) in our data. We train a random forest of $500$ trees using $D_{\text{tr}}$, and report its classification accuracy over $D_{\text{te}}$. This process is repeated ten times at random.

**Discussion of results:** Baseline methods are the lowest performing, up to $59\%$ test accuracy. We believe that the performance of the best baseline, *precedence*, is due to the fact that most Wikipedia is written in the active voice, which often aligns with the temporal sequence of events, and thus correlates with causality. The feature-based methods perform best, achieving up to $85\%$ test classification accuracy. However, feature-based methods enjoy the flexibility of considering each of the $n = 10,000$ elements in the causal projection as a distinct feature. Therefore, feature-based methods do not focus on patterns to be found at a distributional level (such as causality), and are vulnerable to permutation or removal of features. We believe that feature-based methods may achieve their superior performance by exploiting biases in our dataset, which are not necessarily related to causality. As previously mentioned, these numbers are provided only as a **upper-bound** on test performance.

Impressively, the best distribution-based causal discovery method achieves $75\%$ test classification accuracy, which is a significant improvement over the best baseline method. Importantly, our distribution-based methods take a whole 2-dimensional distribution as input to the classifier; as such, these methods are robust with respect to permutations and removals of the $n$ distribution samples. Since the only commonality between training and test instances is their causal structure, and we can solve this problem beyond chance-level, our experiments prove the existence of rich causal structures in raw language data.

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

# Supplementary material to *Causal discovery using proxy variables*

## A  DATASET

The dataset was constructed in two steps: 1) We asked workers from Amazon Mechanical Turk to create pairs of words linked by a causal relation. We provided them with examples of words with a clear causal link (e.g., "sun causes radiation") and examples of related words not sharing a causal relation (e.g., "knife" and "fork"). 2) Each of the pairs collected from the previous step was randomly shuffled and submitted to 20 turks who had not created pairs. Each turk was required to classify the pair of words $(x, y)$ as "$x$ causes $y$", "$y$ causes $x$", or "$x$ and $y$ do not share a causal relation". This procedure resulted in a dataset of $10,000$ causal word pairs $(x, y)$, each accompanied by turk confidence scores.

We now present the detailed instructions given to the AT workers.

### A.1  INSTRUCTIONS FOR WORD PAIR CREATORS

We will ask you to write word pairs (for instance, WordA and WordB) for which you believe the statement "WordA causes WordB" is true.

To provide us with high quality word pairs, we ask you to follow these indications:

- All word pairs must have the form "WordA → WordB". It is essential that the first word (WordA) is the cause, and the second word (WordB) is the effect.

- WordA and WordB must be one word each (no spaces, and no "recessive gene → red hair"). Avoid compound words such as "snow-blind".

- In most situations, you may come up with a word pair that can be justified both as "WordA → WordB" and "WordB → WordA". In such situations, prefer the causal direction with the easiest explanation. For example, consider the word pair "virus → death". Most people would agree that "virus causes death". However, "death causes virus" can be true in some specific scenario (for example, "because of all the deaths in the region, a new family of virus emerged."). However, the explanation "virus causes death" is preferred, because it is more general and depends less on the context.

- We do not accept word pairs with an ambiguous causal relation, such as "book - paper".

- We do not accept simple variations of word pairs. For example, if you wrote down "dog → bark", we will not credit you for other pairs such as "dogs → bark" or "dog → barking".

- Use frequent words (avoid strange words such as "clithridiate").

- Do not rely on our examples, and use your creativity. We are grateful if you come up with diverse word pairs! Please do not add any numbers (for example, "1 - dog → bark"). For your guidance, we provide you examples of word pairs that belong to different categories. Please bear in mind that we will reward your creativity: therefore, focus on providing new word pairs with an evident causal direction, and do not limit yourself to the categories shown below.

**1) Physical phenomenon**: there exists a clear physical mechanism that explains why "WordA → WordB".

- sun → radiation (The sun is a source of radiation. If the sun were not present, then there would be no radiation.)
- altitude → temperature
- winter → cold
- oil → energy

**2) Events and consequences**: WordA is an action or event, and WordB is a consequence of that action or event.

- crime → punishment
- accident → death
- smoking → cancer
- suicide → death
- call → ring

**3) Creator and producer**: WordA is a creator or producer, WordB is the creation of the producer.

- writer → book (the creator is a person)
- painter → painting
- father → son
- dog → bark
- bacteria → sickness
- pen → drawing (the creator is an object)
- chef → food
- instrument → music
- bomb → destruction
- virus → death

**4) Other categories! Up to you, please use your creativity!**

- fear → scream
- age → salary

A.2    INSTRUCTIONS FOR WORD PAIR VALIDATORS

Please classify the relation between pairs of words A and B into one of three categories: either "A causes B", "B causes A", or "Non-causal or unrelated".

For example, given the pair of words "virus and death", the correct answer would be:

- virus causes death (correct);
- death causes virus (wrong);
- non-causal or unrelated (wrong).

Some of the pairs that will be presented are non-causal. This may happen if:

- The words are unrelated, like "toilet and beach".
- The words are related, but there is no clear causal direction. This is the case of "salad and lettuce", since we can eat salad without lettuce, or eat lettuce in a burger.

To provide us with high quality categorization of word pairs, we ask you to follow these indications:

- Prefer the causal direction with the simplest explanation. Most people would agree that "virus causes death". However, "death causes virus" can be true in some specific scenario (for example, "because of all the deaths in the region, a new virus emerged."). However, the explanation "virus causes death" is preferred, because it is true in more general contexts.
- If no direction is clearer, mark the pair as non-causal. Here, conservative is good!
- Think twice before deciding. We will present the pairs in random order!

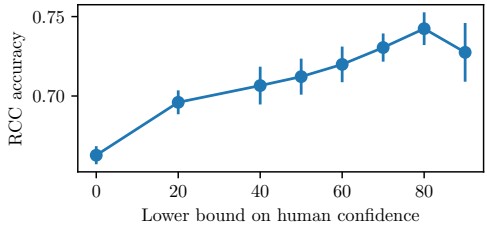

Figure 4: RCC accuracy versus human confidence.

Please classify all the presented pairs. If one or more has not been answered, the whole batch will be invalid. **PLEASE DOUBLE CHECK THAT YOU HAVE ANSWERED ALL 40 WORD PAIRS.**

Examples of causal word pairs:

- "sun and radiation": sun causes radiation
- "energy and oil": oil causes energy
- "punishment and crime": crime causes punishment
- "instrument and music": instrument causes music
- "age and salary": age causes salary

Examples of non-causal word pairs:

- "video and games": non-causal or unrelated
- "husband and wife": non-causal or unrelated
- "salmon and shampoo": non-causal or unrelated
- "knife and gun": non-causal or unrelated
- "sport and soccer": non-causal or unrelated

## B    Relation between RCC and human annotators

Figure 4 shows a positive dependence between the test classification accuracy of RCC and the confidence of human annotations, when considering the test classification accuracy of all the causal pairs annotated with a human confidence of at least $\{0, 20, 40, 50, 60, 70, 80, 90\}$. Thus, our proxy variables and projections arguably capture a notion of causality aligned with the one of human annotators.

## C    Proxy Variables in Machine Learning

The central concept in this paper is the one of *proxy variable*. This is a variable $W$ providing a random source of information related to $x$ and $y$. However, we can consider the reverse process of using a static entity $w$ to augment random statistics about a pair of random variables $X$ and $Y$. As it turns out, this could be an useful process in general prediction problems. To illustrate, consider a supervised learning problem mapping a *feature* random variable $X$ into a *target* random variable $Y$. Such problem is often solved by considering a sample $\{(x_i, y_i)\}_{i=1}^n \sim P^n(X, Y)$. In this scenario, we may contemplate an unpaired, external, *static* source of information $w$ (such as a memory), which might help solving the supervised learning problem at hand. One could incorporate the information in the static source $w$ by constructing the proxy projection $w_i = \pi(x_i, w)$, and add them to the dataset to obtain $\{((x_i, w_i), y_i)\}_{i=1}^n$ to build the predictor $f(x, \pi(x, w))$.

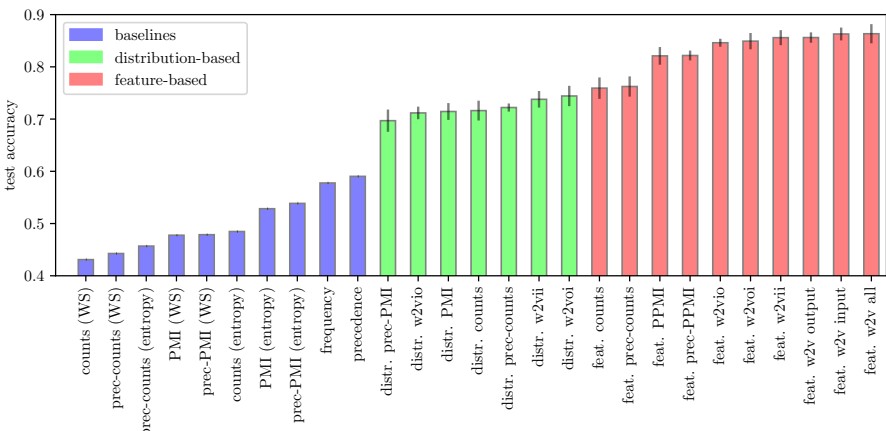

Figure 5: Results for all methods on the NLP experiment. Accuracies above $52\%$ are statistically significant with respect to a Binomial test at a significance level $\alpha = 0.05$.

## D  EXPERIMENTS: EXTENDED RESULTS

In this section, we compare more methods from the three families introduced in Section 4. The entended results are found in Figure 5.

Our computations are based on the full English Wikipedia, as post-processed by Matt Mahoney (see `http://www.mattmahoney.net/dc/textdata.html`). We study the $N = 1,970$ pairs of words out of $10,000$ from the dataset described in Appendix A that achieved a consensus across at least $18$ out of $20$ turks. Since turk confidence is used to measure the strength of the causal association, we consider for these pairs with strong confidence that the problem is binary (i.e., we ignore the possibility that the words in the pair are related but not causal). We use RCC to estimate the causal relation between pairs of random variables.

### D.1  PROXY PROJECTIONS

Throughout our experimental evaluation, we will use and compare different proxy projections:

1) $\pi_{\text{w2vii}}(w, x) = \langle v_w^i, v_x^i \rangle$, where $v_z^i \in \mathbb{R}^d$ is the *input* word2vec representation Mikolov et al. (2013) of the word $z$. The dot-product $\langle v_w^i, v_x^i \rangle$ measures the *similarity in meaning* between $w$ and $x$.

2) $\pi_{\text{w2vio}}(w, x) = \langle v_w^i, v_x^o \rangle$, where $v_z^o \in \mathbb{R}^d$ is the *output* word2vec representation of the word $z$. The dot-product $\langle v_w^i, v_x^o \rangle$ is an unnormalized estimate of the conditional probability $p(x|w)$ Melamud et al. (2015).

3) $\pi_{\text{w2voi}}(w, x) = \langle v_w^o, v_x^i \rangle$, an unnormalized estimate of the conditional probability $p(w|x)$.

4) $\pi_{\text{counts}}(w, x) = p(w, x)$, where the pmf $p(w, x)$ is directly estimated from counting within-sentence co-occurrences in the corpus.

5) $\pi_{\text{prec-counts}}(w, x)$ similar to the one above, but only over sentences where $w$ precedes $x$.

6) $\pi_{\text{pmi}}(w, x) = p(w, x)/(p(w)p(x))$, where the pmfs $p(w)$, $p(x)$, and $p(w, x)$ are estimated from counting words and (sentence-based) co-occurrences in the corpus. The log of this quantity is known as point-wise mutual information, or PMI Church & Hanks (1990).

7) $\pi_{\text{prec-pmi}}(w, x)$, similar to the one above, but only over sentences where $w$ precedes $x$.

Applying the causal projections to our sample from proxy $W$, we construct the $n$-vector

$$\Pi_{\text{proj}}^x = (\pi_{\text{proj}}(w_1, x), \ldots, \pi_{\text{proj}}(w_n, x)), \tag{1}$$

and similarly for $\Pi_{\text{proj}}^y$, where proj $\in \{$w2vii, w2vio, w2voi, counts, prec-counts, pmi, prec-pmi$\}$. We use the skip-gram model of fastText Bojanowski et al. (2016) to compute $300-$dimensional word2vec representations.

## D.2    BASELINES

These are a variety of unsupervised baselines. Each baseline computes two scores, denoted by $S_{x \to y}$ and $S_{x \leftarrow y}$, predicting $x \to y$ if $S_{x \to y} > S_{x \leftarrow y}$, and $x \leftarrow y$ if $S_{x \to y} < S_{x \leftarrow y}$. The baselines are: i) *frequency*: $S_{x \to y}$ is the number of sentences where $x$ appears in the corpus, and $S_{x \leftarrow y}$ is the number of sentences where $y$ appears in the corpus, ii) *precedence*: considering only sentences from the corpus where both $x$ and $y$ appear, $S_{x \to y}$ is the number of sentences where $x$ occurs before $y$, and $S_{x \leftarrow y}$ is the number of sentences where $y$ occurs before $x$, iii) *counts (entropy)*: $S_{x \to y}$ is the entropy of $\Pi^x_{\text{counts}}$, and $S_{x \leftarrow y}$ is the entropy of $\Pi^y_{\text{counts}}$, as defined in (1), iv) *counts (WS)*: Using the WS measure of Weeds & Weir (2003), $S_{x \to y} = \text{WS}(\Pi^x_{\text{counts}}, \Pi^y_{\text{counts}})$, and $S_{x \leftarrow y} = \text{WS}(\Pi^y_{\text{counts}}, \Pi^x_{\text{counts}})$, v) *prec-counts (entropy)*: $S_{x \to y}$ is the entropy of $\Pi^x_{\text{prec-counts}}$, and $S_{x \leftarrow y}$ is the entropy of $\Pi^y_{\text{prec-counts}}$ (1) and vi) *prec-counts (WS)*: analogous to the previous.

