# OpenReview forum: "Causal Discovery Using Proxy Variables"
_ICLR.cc/2018/Workshop — Accept_

### Official Review · AnonReviewer2 · 2018-03-09
**Are you sure the relation is causality**

**Rating:** 7
**Confidence:** 4

**Review:**

The paper aims to discover relations between two objects and seems to be a very solid paper. Experiments are carried out on two different types of datasets: image-based and linguistic-based.

Yet, it is unclear the reason why the relations between two objects are called causal relations. In fact, the relation between the two images does not seem to be related to the notion of causality. The original image does not seems to be "cause" of the modified image. It seems to be only related. Can you be more specific and convincing?

---

### Official Review · AnonReviewer3 · 2018-03-10
**Interesting application, a bit unclear theory, promising but limited experiments**

**Rating:** 6
**Confidence:** 4

**Review:**

The paper discusses the application of cause-effect methods to the case of static entities (e.g. images or words). Since these methods work on random variables, the main idea of the paper is how to construct a random variable for a static entity using a proxy variable. Given the random variables representing two static entities, one can apply the existing cause-effect methods (e.g. ANM) to discover the causal relation between the entities. This idea is applied to three use cases: finding the original vs transformed image, finding the order of frames in a video and discovering the cause-effect relation between concepts represented as words, e.g. “virus” causes “death”.

The paper seems novel and has some interesting applications of causal discovery. I found it a bit hard to read (also given the three page limit), so I resorted to a longer version (https://arxiv.org/abs/1702.07306) to understand some of the details. I have some concerns regarding the theoretical framework (there is not real cause-effect relation between the observed entities), its general applicability and the evaluation in the word pairs (some of the non causal methods seems to outperform the causal ones, although the authors present some possible explanations). On the other hand, maybe the ICLR workshop is a place in which some of these concerns could be discussed.

Pros:
- Interesting application of cause-effect methods, could open the area to many new applications (e.g. plagiarism detection, image reuse, etc.).
- Very good results on detecting which image is the original, and which video frame “causes” another video frame.

Cons:
- Unclear theory, the random entities don’t really cause each other, so it’s a bit unclear to me what are the cause-effect methods really discovering.
- The promising experiments on images and videos are limited in scope (one dataset of DeepArt transformations of images and one single video example), which makes it unclear whether the results would generalize.
- I don’t completely understand what is a causal relation between words. I assume it is the causal relation between the underlying concepts, but trying to find it from text one may get instead the causal relation between having one word present in a text and another word in the same text.

---

### Decision · Program_Chairs · 2018-03-20
**ICLR 2018 Workshop Acceptance Decision**

**Decision:**

Accept

**Comment:**

Congratulations, your paper was accepted to the ICLR workshop.